# Fertility return after hormonal contraceptive discontinuation and associated factors among women attended Family Guidance Association of Ethiopia Dessie model clinic, Northeast Ethiopia: A cross-sectional study

**Yitayish Damtie**[1]*, **Bereket Kefale**[2], **Mastewal Arefaynie**[2], **Melaku Yalew**[1], **Bezawit Adane**[1]

1 Department of Public Health, College of Medicine and Health Sciences, Injibara University, Injibara, Ethiopia, 2 Department of Reproductive and Family Health, School of Public Health, College of Medicine and Health Sciences, Wollo University, Dessie, Ethiopia

* yitutile@gmail.com

## Abstract

### Background

Women who use hormonal contraception face delayed return of fertility upon discontinuation. There was limited evidence of fertility return after hormonal contraceptive discontinuation in the study area. Hence this study assessed fertility return after hormonal contraceptive discontinuation and associated factors among pregnant women attending Family Guidance Association Ethiopia (FGAE) Dessie model clinic, Northeast Ethiopia, 2019.

### Methods

A cross-sectional study was conducted on 423 samples selected by using systematic random sampling. Data were collected by face-to-face interview using a pretested and structured questionnaire and reviewing client records. Data were entered using Epi Data version 3.1 and analyzed using SPSS version 23. Both bi-variable and multivariable binary logistic regressions were used to identify predictors of delayed fertility return. Adjusted odds ratio (AOR) along with a 95% Confidence Interval (CI) was used to measure the strength and the direction of the association and statistical significance was declared at a P-value less than 0.05.

### Result

The proportion of fertility return among currently pregnant women after discontinuation of any hormonal contraceptive methods was 88.6% (95% CI; (85.6%-92%)). The proportion of fertility return among Depo-Provera, implant, Intrauterine Contraceptive Device (IUCD), and Oral Contraceptive Pill (OCP) users was 75%, 99.1%, 100%, and 97.8% respectively. Age,

**Data Availability Statement:** All relevant data are within the paper and its Supporting Information files.

**Funding:** The author(s) received no specific funding for this work.

**Competing interests:** The authors have declared that no competing interests exist.

**Abbreviations:** ANC, Antenatal Care; AOR, Adjusted odds ratio; CI, Confidence Interval; FGAE, Family Guidance Association of Ethiopia; IQR, Inter Quartile Range; IUCD, Uterine Contraceptive Device; OCP, Oral Contraceptive Pill; STI, Sexual Transmitted Infection; SSA, Sub-Saharan Africa.

(AOR = 5.37, (95% CI; (1.48, 13.6)) and using Depo-Provera (AOR = 4.82, 95% CI; (1.89, 14.2)) had a significant association with delayed fertility return.

## Conclusions

The proportion of fertility return among women after discontinuation of any hormonal contraceptive methods was high. Age and using Depo-Provera had a positive association with delayed fertility return. This study recommends a contraceptive counseling approach that addresses concerns about delay in the return of fertility after hormonal contraceptive discontinuation to avoid confusion among family planning users.

## Introduction

A family planning program is an essential and cost-effective health promotion program that has been provided in the African context for more than 50 years [1]. Universal access to family planning can reduce maternal deaths by 40% and infant mortality by 10% [2, 3]. Despite much programmatic success all over the world, only 22% of reproductive-age women used modern contraception, and 17.3% of women had an unmet need for family planning in 2019 [4]. Unmet need for family planning is also high in Sub-saran Africa (SSA) with an estimated 23.5% of women who want to limit or space their births are not using any contraceptive method [5]. In Ethiopia, only 36% of currently married women used contraceptives and 22% of women have an unmet need for family planning [6]. Lack of awareness about contraceptive methods, lack of access to contraceptive information, fear of infertility, and cost of the services are barriers responsible for the huge unmet need for family planning among individuals [5, 7–9].

Delayed return of fertility after discontinuation of contraception becomes a big challenge for women who are using hormonal contraception [10–15]. Although hormonal contraceptive methods are effective, safe, and reversible, they delay fertility upon discontinuation [16–18]. A study showed that 25%, 28%, 25%, and 36% of previous OCP, ICUD, implants, and injectable users experienced a delay in the return of fertility within one year of contraceptive discontinuation [19]. Another study conducted in fifteen SSA countries indicated that 27% of women were unable to become pregnant within a year following the discontinuation of hormonal contraception [20]. Different evidence suggested that Post-pill amenorrhea was observed among previous OCP users [21–25].

Delay in return of fertility after hormonal contraceptive discontinuation had a significant impact on women's health. It is a repeatedly mentioned reason for not using contraception [26, 27]. It leads to early contraceptive method discontinuation and dissatisfaction with family planning services [28]. The delay in return of fertility following hormonal contraceptive discontinuation is also linked with stigma and discrimination, isolation, intimate partner violence, and mental health disorders [29–31].

Studies showed that different factors have been associated with delayed fertility return. These include age [20, 32], smoking, alcohol drinking [33], parity, gravidity [32, 34], duration of contraceptive use, and type of contraceptive method [15, 35].

There was limited evidence regarding fertility return after hormonal contraceptive discontinuation in the study area. So, this study aimed to assess fertility return after hormonal contraceptive discontinuation and associated factors among pregnant women attending FGAE Dessie Model Clinic. The finding of this study will have paramount importance for

policymakers and program designers to design evidence-based interventions to increase the utilization of family planning services.

## Material and methods

### Study area, study design, and participants

An institution-based cross-sectional study was conducted in FGAE Dessie Model Clinic from May 1–30/2019. FGAE is a volunteer-based, non-government and nonprofit organization that initiate and expand the Sexual and Reproductive Health (SRH) program in Ethiopia since 1966. FGAE Dessie Model clinic which is found in Dessie city administration (located 401KM away from Addis Ababa, the capital city of Ethiopia, and 480km away from Bahir Dar) is one of FGAE SRH clinics established in 1975 and has served more than 100,000 population of all types including key and priority population, most vulnerable and underserved populations of South Wollo administrative zone. It is one of the local Non-governmental Organizations (NGOs) that provide SRH services including youth-friendly services, family planning, antenatal, delivery, postnatal, HIV, and other STI services through its clinic.

The source population was all pregnant women attending the Antenatal Care (ANC) unit of FGAE Dessie Model Clinic whereas; the study population was all systematically selected pregnant women attending the ANC unit of FGAE Dessie Model Clinic during the study period. All systematically selected pregnant women coming for ANC during the data collection period were included and those women who became pregnant without using any contraceptive method and due to contraceptive failure were excluded from the study.

### Sample size and sampling procedure

The sample size was determined by using single population proportion formula by considering the proportion of delayed fertility return as 50% since no study was done in Ethiopia, 95% confidence level, and a 5% margin of error. Thus, the final sample size after adding a 10% non-response rate became 423. A systematic random sampling technique was used to select study participants. As the 2018 data showed, on average, a total of 936 pregnant women attended the ANC room of FGAE Dessie model clinic each month. Using this information, interval (k) was determined by dividing the average client flow per month (N) by the total sample size (n) i.e. k = N/n, K = 936/423, K = 2.2≈2. So, study participants were selected every two women until reaching the final sample size.

### Data collection procedures and measurements

Data were collected by face-to-face interview using a pretested and structured Amharic version questionnaire taken from previous similar works of literature. The questionnaire was composed of Socio-demographic factors, behavioral factors, obstetric, contraceptive, disease, and nutritional-related factors [15, 20, 32–36]. Two trained nurses have collected the data from May 1–30/2019 under the supportive supervision of one supervisor and principal investigators. Special markings were used to avoid the collection of unnecessary data from cases with repeated visits during the study period.

Data collectors and supervisor were trained for two days on the objective of the study, the content of the questionnaire, and the data collection procedure. Before the data collection, the questionnaire was pretested on 22 study participants at Dessie health center, and based on feedback obtained from the pretest, a necessary modification was done. During the study period, the collected data were checked continuously daily for completeness by the supervisor and principal investigators. Moreover, we have tried to provide adequate time and detailed

explanation of questioners for each study participant to minimize memory-related bias (recall bias).

The outcome variable was fertility return after hormonal contraceptive discontinuation. In this study, fertility return was defined if conception occurred within 12 months following discontinuation of any hormonal contraceptive methods (Depo-Provera, OCP, IUCD, and implant) [15, 36].

**Abortion**: Termination of pregnancy before 28 weeks of gestation. It includes both spontaneous and induced abortion.

**The menstrual cycle** is the duration between two consecutive menses (the duration from the first day of menses to the first day of the next menses). The cycle is said to be irregular: if a woman's menstrual cycle is shorter than 21 days or longer than 35 days and, regular: if a cycle ranges from 21 to 35 days.

**History of medical illness**: Having a previous history of tuberculosis, diabetes, or both.

**Khat** is a fresh green leafy plant that contains a psycho-active ingredient cathinone. It is classified as an illicit substance due to the potential for psychological dependence. Its regular consumption negatively impacts the human central nervous system, systemic blood pressure, psychological health, and reproductive system causing reproductive toxicity and sexual dysfunction.

## Statistical analysis

Data were coded and entered into Epi Data version 3.1 and exported to SPSS version 23 for analysis. Descriptive statistics such as frequency, percentage, and median with Interquartile Range (IQR) were carried out. Bi-variable binary logistic regression was performed and variables with a p-value of less than 0.25 were transported to multivariable logistic regression. Multicollinearity was checked using standard error and Hosmer and Lemeshow goodness of test was used to check model fitness. Variables with a P-value less than 0.05 and AOR with a 95% confidence interval non-inclusive of one were considered as statistically significant predictors of delayed fertility return in the final model.

## Ethical approval

Ethical Clearance was taken from the Ethical Review Committee (ERC) of Wollo University College of Medicine and Health Sciences. An official letter was written from the School of Public Health to FGAE Dessie Model Clinic head to get permission. After explaining the purpose of the study, verbal informed consent was taken from each participant before the data collection. They were informed that participating in the study was voluntary and the right to withdraw from the study at any time during the interview was assured. Privacy and confidentiality of information they gave was secured at all levels.

This manuscript was organized and written according to strengthening the Reporting of Observational Studies in Epidemiology (STROBE) 2007 (v4) Statement checklist for cross-sectional studies (S1 Table).

## Results

### Socio-demographic characteristics

In this study, a total of 402 pregnant women were involved and making a response rate of 94.5%. The median age of the respondents was 28 years with an IQR of 4 years. Three hundred eighty (94.5%) of pregnant women were married and 361(89.8%) of pregnant women were Amhara in their ethnicity respectively. Two hundred sixteen (53.7%) pregnant women were

**Table 1. Socio-demographic characteristics of pregnant women who attended FGAE Dessie model clinic, Northeast Ethiopia, 2019.**

| Variable | Frequency (n = 402) | Percentage (%) |
|---|---|---|
| **Age** | | |
| 15–24 | 78 | 19.4 |
| 25–34 | 295 | 73.4 |
| ≥35 | 29 | 7.2 |
| **Marital status** | | |
| Single | 10 | 2.5 |
| Married | 380 | 94.5 |
| Divorced | 12 | 3.0 |
| **Ethnicity** | | |
| Amhara | 361 | 89.8 |
| Oromo | 23 | 5.7 |
| Tigray | 18 | 4.5 |
| **Income** | | |
| ≤94.3$ | 203 | 50.5 |
| 94.4–188.7$ | 163 | 40.5 |
| >188.8$ | 36 | 9.0 |
| **Religion** | | |
| Orthodox tewahido | 216 | 53.7 |
| Muslim | 175 | 43.5 |
| Protestant | 11 | 2.7 |
| **Residence** | | |
| Urban | 392 | 97.5 |
| Rural | 10 | 2.5 |
| **Educational status** | | |
| No formal education | 17 | 4.2 |
| Grade 1–8 | 65 | 16.2 |
| Grade 9–12 | 124 | 30.8 |
| College and above | 196 | 48.8 |
| **Occupation** | | |
| Government employ | 143 | 35.6 |
| Private employ | 48 | 11.9 |
| Housewife | 149 | 37.1 |
| Merchant | 54 | 13.4 |
| Farmer | 3 | .7 |
| Daily laborer | 5 | 1.2 |

orthodox tewahido followers, 203(50.5%) of women earn less than or equal to 94.3 American Dollars, and 392 (97.5%) of women live in urban areas respectively. One hundred ninety-six (48.8%) of pregnant women were educated up to college and university level and 143(35.6%) of women were government employed respectively (Table 1).

## Behavioral characteristics

One hundred thirty-one (32.6%) of pregnant women ever drink alcohol in their lifetime, 121 (30.1%) of women drink alcohol in the last twelve months and 64 (15.9%) of women drink alcohol less than once a month in the last twelve months respectively. Seventy-one (17.7%) of pregnant women ever chew khat in their lifetime, 51(12.7%) of women chewed khat in the last

**Table 2. Behavioral characteristics of pregnant women who attended FGAE Dessie model clinic, Northeast Ethiopia, 2019.**

| Variables | Frequency (n = 402) | Percentage (%) |
|---|---|---|
| **Ever drink alcohol** | | |
| Yes | 131 | 32.6 |
| No | 271 | 67.4 |
| **Alcohol use within 12 months** | | |
| Yes | 121 | 30.1 |
| No | 10 | 2.5 |
| **Frequency of alcohol use** | | |
| Daily | 1 | 0.2 |
| One to four days per week | 3 | 0.7 |
| One to three days per month | 53 | 13.2 |
| Less than once a month | 64 | 15.9 |
| **Ever chew khat** | | |
| Yes | 71 | 17.7 |
| No | 331 | 82.3 |
| **Chew khat within 12 months** | | |
| Yes | 51 | 12.7 |
| No | 20 | 5.0 |
| **Frequency of chewing** | | |
| Monthly | 5 | 1.2 |
| Less than once a month | 17 | 4.2 |
| Rarely | 29 | 7.2 |
| **Frequency of sexual intercourse** | | |
| Three times a day | 71 | 17.7 |
| Three times a week | 219 | 54.5 |
| Three times a month | 102 | 25.4 |
| Three times a year | 10 | 2.5 |

twelve months and 17 (4.2%) of women chew khat less than once a month in the last twelve months respectively. In the case of frequency of sexual intercourse, 71 (17.7%) and 219 (54.5%) of pregnant women had sexual intercourse three times a day and three times a week respectively (Table 2).

## Contraceptive, obstetric, and disease-related characteristics

The majority 172(42.8%) of pregnant women used Depo-Provera and 113 (28.1%) women used implants as a contraceptive method before the current pregnancy. One hundred eighty-four (45.8%) of pregnant women used contraceptive methods for 12–24 months before the current pregnancy. One hundred fifty-eight (39.3%) of women have given birth once, 30 (7.5%) of pregnant women had a history of abortion, and 301(74.9%) of pregnant women had regular menstrual cycles before the current pregnancy. Four (1%) and 6(1.5%) of pregnant women had a history of Sexual Transmitted Infection (STI) and other medical illnesses (tuberculosis and diabetic Malthus) respectively (Table 3).

## The proportion of fertility return

The proportion of fertility return among pregnant women after discontinuation of any hormonal contraceptive methods before the current pregnancy was 88.6% (95% CI; (85.6–92%)).

**Table 3. Contraceptive, obstetric, and disease-related characteristics of pregnant women attended FGA Dessie model clinic, Northeast Ethiopia, 2019.**

| Variables | Frequency (n = 402) | Percentage (%) |
|---|---|---|
| **Type of contraceptive used** | | |
| OCP | 91 | 22.6 |
| Depo-Provera | 172 | 42.8 |
| Implant | 113 | 28.1 |
| IUCD | 26 | 6.5 |
| **Duration of contraceptive use** | | |
| <12 month | 107 | 26.6 |
| 12–24 month | 184 | 45.8 |
| 25–36 month | 63 | 15.7 |
| >36 month | 48 | 11.9 |
| **Number of births** | | |
| Never give birth | 92 | 22.9 |
| 1 | 158 | 39.3 |
| ≥2 | 152 | 37.8 |
| **History of abortion** | | |
| Yes | 30 | 7.5 |
| No | 372 | 92.5 |
| **Menstrual cycle** | | |
| Irregular | 101 | 25.1 |
| Regular | 301 | 74.9 |
| **History of STI** | | |
| Yes | 4 | 1.0 |
| No | 398 | 99.0 |
| **History of medical illness** | | |
| Yes | 6 | 1.5 |
| No | 396 | 98.5 |

The proportion of fertility return among pregnant women who used Depo-Provera, implant, IUCD, and OCP before the current pregnancy was 75% (95% CI; (68.6%-80.8%)), 99.1% (95% CI; (97.3–99.999%)), 100% and 97.8% respectively.

The median time of fertility return among pregnant women after discontinuation of any hormonal contraceptive methods before the current pregnancy was 6 months with an IQR of 8 months. The median time of fertility return among Depo-Provera implant, IUCD, and OCP users before the current pregnancy was 9 months, 4 months, 6 months, and 2 months with IQR of 7 months, 4 months, 8 months, and 4 months respectively.

## Factors associated with delayed fertility return

Both bi-variable and multivariable binary logistic regression analyses were done. The finding indicated that pregnant women aged 35 years and more were 5.4 times more likely to experience fertility delay upon hormonal contraceptive discontinuation compared to their counterparts (AOR = 5.37, (95% CI; (1.48, 13.6)). Similarly, pregnant women who used Depo-Provera before the current pregnancy were 4.8 times more likely to experience fertility delay upon discontinuation as compared to women who used implant, IUCD, and OCP (AOR = 4.82, 95% CI; (1.89, 14.2)) (Table 4).

**Table 4. Factors associated with delayed fertility return after hormonal contraceptive discontinuation among pregnant women attended FGAE Dessie model clinic, Northeast Ethiopia, 2019.**

| Variables | Fertility return (n = 402) | | COR (95% CI) | AOR (95% CI) |
|---|---|---|---|---|
| | Delayed | Not delayed | | |
| **Age** | | | | |
| 15–24 | 6 (13%) | 72 (20.2%) | 1 | **1** |
| 25–34 | 33 (71.7%) | 262 (73.6%) | 1.51 (0.61, 3.75) | 1.35 (0.44, 4.15) |
| ≥35 | 7 (15.2%) | 22 (6.2%) | 3.82 (1.16, 12.6) | 5.37 (1.48, 13.6)* |
| **Monthly Income** | | | | |
| ≤94.3$ | 26 (56.5%) | 177 (49.7%) | 1 | 1 |
| 94.4–188.7$ | 17 (37.0%) | 146 (41.0%) | 0.79 (0.41, 1.52) | 0.90 (0.39, 2.04) |
| >188.8$ | 3 (6.5%) | 33 (9.3%) | 0.62 (0.18, 2.16) | 1.18 (0.24, 5.75) |
| **Educational status** | | | | |
| No formal education | 6 (13.0%) | 11 (3.1%) | 4.55(1.52, 13.6)* | 2.63 (0.41, 16.94) |
| Grade 1–8 | 8 (17.4%) | 57 (16.0%) | 1.17 (0.49, 2.8) | 1.27 (0.31, 5.21) |
| Grade 9–12 | 11 (23.9%) | 113 (31.7%) | 0.81 (0.38, 1.7) | 0.48 (0.14, 1.57) |
| College and above | 21 (45.7%) | 175 (49.2%) | 1 | 1 |
| **Ever drink alcohol** | | | | |
| Yes | 15 (32.6%) | 116 (32.6%) | 1.01 (0.52, 1.9) | 1.14 (0.48, 2.72) |
| No | 31 (67.4%) | 240 (67.4%) | 1 | 1 |
| **Ever chew Khat** | | | | |
| Yes | 11 (23.9%) | 60 (16.9%) | 1.55 (0.75, 3.2) | 0.99 (0.27, 3.62) |
| No | 35 (76.1%) | 296 (83.1%) | 1 | 1 |
| **Duration of contraceptive use** | | | | |
| <12 month | 10 (21.7%) | 97 (27.2%) | 1 | 1 |
| 12–24 month | 23 (50.0%) | 161 (45.2%) | 1.39 (0.63, 3.04) | 1.02 (0.39, 2.62) |
| 25–36 month | 6 (13.0%) | 57 (16.0%) | 1.02 (0.35, 2.96) | 1.28 (0.36, 4.49) |
| >36 month | 7 (15.2%) | 41 (11.5%) | 1.66 (0.59, 4.65) | 1.10 (0.28, 4.36) |
| **Number of births** | | | | |
| Never give birth | 9 (19.6%) | 83 (23.3%) | 0.81 (0.35, 1.88) | 1.67 (0.49, 5.61) |
| 1 | 19 (41.3%) | 139 (39%) | 1.02 (0.51, 2.02) | 1.19 (0.47, 3.01) |
| ≥2 | 18 (39.1%) | 134 (37.6%) | 1 | 1 |
| **Menstrual cycle** | | | | |
| Irregular | 19 (41.3%) | 82 (23.0%) | 2.4 (1.24,4.44)** | 1.21 (0.53, 2.75) |
| Regular | 27 (58.7%) | 274 (77.0%) | 1 | 1 |
| **MUAC** | 46 (11.4%) | 356(88.6%) | 0.92 (0.79, 1.07) | 0.99 (0.82, 1.20) |
| **Occupation** | | | | |
| Government employ | 17 (37.0%) | 126 (35.4%) | 1 | 1 |
| Private employ | 4 (8.7%) | 44 (12.4%) | 0.67 (0.22, 2.11) | 0.73 (0.19, 2.87) |
| Housewife | 14 (30.4%) | 135 (37.9%) | 0.77 (0.36, 1.62) | 0.47 (0.13, 1.74) |
| Others[c] | 11 (23.9%) | 51 (14.3%) | 1.60 (0.70, 3.65) | 0.98 (0.27, 3.67) |
| **Frequency of sexual intercourse** | | | | |
| Three times a day | 10 (21.7%) | 61 (17.1%) | 1 | 1 |
| Three times a week | 24 (52.2%) | 195 (54.8%) | 0.751(0.34,1.68) | 0.65 (0.18, 2.33) |
| Others[d] | 12 (26.1%) | 100 (28.1%) | 0.732(0.34,1.80) | 0.48 (0.11, 2.09) |
| **Type of contraceptive** | | | | |
| Depo-provera | 43 (93.5%) | 129 (36.2%) | 3.21 (1.67, 13.3)*** | 4.82 (1.89, 14.2)*** |

*(Continued)*

**Table 4.** (Continued)

| Variables | Fertility return (n = 402) | | COR (95% CI) | AOR (95% CI) |
|---|---|---|---|---|
| | Delayed | Not delayed | | |
| Others [e] | 3(6.5%) | 227(63.8%) | 1 | 1 |

COR, crude odds ratio; AOR, adjusted odds ratio; MUAC, middle-upper arm circumference

*significant at P<0.05

** significant at P<0.01

*** significant at P ≤ 0.001 in the bi-variable and multivariable logistic regression analysis

[c] merchant, farmer, and daily laborer

[d] three times a month and three times a year

[e] IUCD and OCP.

## Discussion

The proportion of fertility return after discontinuation of any hormonal contraceptive methods before the current pregnancy was 88.6%. The proportion of fertility return among Depo-Provera, implant, IUCD, and OCP users before the current pregnancy was 75%, 99.1%, 100%, and 97.8% respectively. Age and using Depo-Provera had a positive association with delayed fertility return.

The proportion of fertility return among pregnant women after discontinuation of any hormonal contraceptive methods was 88.6%. the finding is similar to a study conducted by *Farrow A et al* in England which showed that 82% of the participants conceived within one year after discontinuation of any hormonal contraceptive method [36]. However, it is higher than studies conducted by *Barden-O'Fallon et al* in fifteen SSA countries and *Girum T et al* which is 73% and 83.1% respectively [15, 20].

The proportion of fertility return among Depo-Provera users before the current pregnancy was similar but, the proportion of fertility return among implant, IUCD, and OCP users was high as compared to a meta-analysis study conducted by *Girum T et al* which indicated that the proportion of fertility return among Depo-Provera, implant, IUCD, and OCP users was 77.7%, 74.7%, 84.7%, and 88% [15]. Similarly, the proportion of fertility return among Depo-Provera, implant, IUCD, and OCP users was high as compared to a study conducted by *Gayatri M et al* in Indonesia [37]. The proportion of fertility return among IUCD users was also high as compared to a study conducted by *Stoddard AM et al* in the United States of America and *Tadesse E* which was 81% and 86.1% respectively [34, 38]. The discrepancy could be due to differences in the study setting, study period, and study population.

In this study, age has a significant association with delayed fertility return. The finding is similar to studies conducted by *Farrow A et al*, *Buckshee K et al*, *Sivin I et al and Barden-O'Fallon et al* [20, 36, 39, 40]. The reason behind this could be the number of eggs decreases as women get older due to a fixed number of eggs in the ovary. The other reason could be as age increases, women will be at higher risk of disorders that can affect fertility, such as uterine fibroids and endometriosis. Moreover, as age increases, the remaining eggs in older women are more likely to have abnormal chromosomes [41, 42].

Using Depo-Provera had a positive association with delayed fertility return. A global handbook on family planning for healthcare providers demonstrated that Depo-Provera causes a delayed fertility return [31]. The finding is also consistent with studies conducted by *Yland JJ et al* and *Pardthaisong T et al* [43, 44], This could be due to Depo-Provera can stay in the body system longer than the other birth control methods so that the clearance of progestin from the serum takes a long time and the meantime to ovulation become delayed. On the other hand,

Depo-Provera causes excessive weight gain; these leads women to stop ovulation and get irregular menstrual cycles. Once women stop ovulation, they would not be able to conceive and therefore it delays fertility return.

This study has limitations. Some variables like a history of alcohol use, chat chewing, duration of contraceptive use, sexual behavior, and duration of hormonal contraceptive discontinuation until the current pregnancy will be affected by recall bias. Being a facility-based study will also underestimate fertility return as it is prone to miss defaulters and delayed visitors to their ANC appointment.

## Conclusion

The proportion of fertility return among pregnant women after discontinuation of any hormonal contraceptive methods before the current pregnancy was high. Age and using Depo-Provera had a positive association with delayed fertility return. The Ministry of Health should design a contraceptive counseling approach that addresses concerns about delays in the return of fertility after contraceptive discontinuation to avoid confusion among family planning users.

## Supporting information

**S1 Table. STROBE checklist.**
(DOCX)

**S1 Dataset. The data set used to assess fertility return after hormonal contraceptive discontinuation and associated factors among pregnant women attending Family Guidance Association Ethiopia (FGAE) Dessie model clinic.**
(SAV)

## Acknowledgments

We would like to extend our appreciation to FGA Dessie Model Clinic Head and staff, study participants, data collectors, and supervisor for their cooperation during the data collection process.

## Author Contributions

**Conceptualization:** Yitayish Damtie.

**Data curation:** Yitayish Damtie, Mastewal Arefaynie, Melaku Yalew.

**Formal analysis:** Yitayish Damtie, Bereket Kefale.

**Investigation:** Yitayish Damtie, Bereket Kefale, Melaku Yalew, Bezawit Adane.

**Methodology:** Yitayish Damtie, Bereket Kefale, Mastewal Arefaynie, Bezawit Adane.

**Software:** Yitayish Damtie, Bereket Kefale, Bezawit Adane.

**Supervision:** Mastewal Arefaynie, Melaku Yalew.

**Writing – original draft:** Yitayish Damtie.

**Writing – review & editing:** Yitayish Damtie, Mastewal Arefaynie, Melaku Yalew, Bezawit Adane.

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
