## [Decision Letter · Decision Letter 0]

22 Feb 2022

PONE-D-21-25440Fertility return after contraceptive discontinuation and associated factors among women attended Family Guidance Association of Dessie model clinic, Northeast Ethiopia: a cross-sectional studyPLOS ONE

Dear Dr. Damtie,

Thank you for submitting your manuscript to PLOS ONE. After careful consideration, we feel that it has merit but does not fully meet PLOS ONE’s publication criteria as it currently stands. Therefore, we invite you to submit a revised version of the manuscript that addresses the points raised during the review process.

 Please see the attached comments from the reviewers, who gave detailed advice.

We look forward to receiving your revised manuscript.

Kind regards,

Janet E Rosenbaum, Ph.D.

Academic Editor

PLOS ONE

https://journals.plos.org/plosone/s/fileid=ba62/PLOSOne_formatting_sample_title_authors_affiliations.pdf".

2. Please amend your current ethics statement to address the following concern: Please explain i) why written consent was not obtained, ii) how you documented participant consent, and iii) whether the ethics committees/IRB approved this consent procedure.

Reviewers' comments:

Reviewer's Responses to Questions

**Comments to the Author**

1. Is the manuscript technically sound, and do the data support the conclusions?

Reviewer #1: Partly

Reviewer #2: Partly

Reviewer #3: Partly

2. Has the statistical analysis been performed appropriately and rigorously? 

Reviewer #1: I Don't Know

Reviewer #2: Yes

Reviewer #3: I Don't Know

3. Have the authors made all data underlying the findings in their manuscript fully available?

Reviewer #1: No

Reviewer #2: Yes

Reviewer #3: No

4. Is the manuscript presented in an intelligible fashion and written in standard English?

Reviewer #1: No

Reviewer #2: Yes

Reviewer #3: Yes

5. Review Comments to the Author

Reviewer #1: Comments

1. Introduction ..the first paragraph is unnecessary( It talks about infertility and....).

2.The fourth paragraph is unnecessary...The introduction should focus on fertility return after contraceptive discontinuation and associated factors ..but I didn't see about associated factors

3. Methods: How do you control reputation of cases. The same patient might came twice per week or weekly.

4. I wonder that you get more than 400 pregnant mothers who fulfills the study criteria within a month (even less than 22 days in one clinic). I have a question on the reliability of the data.

5.There were ampoule studies related to your title in Ethiopia.. please search more. E.g.[https://pubmed.ncbi.nlm.nih.gov/8698014/)

6. This study had a lot of confounders especially memory related, sexually behavior to get pregnant.[ They may discontinued due to side effects or other different reasons.] What was done to minimize thus confounders?

7. What does mean 'unit increase in age...'

8. What is your base to define delayed fertility after 12 months. Or operational definition?

9. Discussion is superficial

10. conclusion: Is so elementary...should be improved.

Reviewer #2: Title: the title “Fertility return after contraceptive discontinuation and associated factors

among women” is about fertility return after discontinuation, however, what I presented in the result is proportion if delayed fertility, which is the other way around. So it is better described as per the research topic and the objectives

Abstract

Result: the phrase “Uterine contraceptive device has to be replaced by Intrauterine contraceptive device”

The percentages have to be out of the parenthesis.

The conclusion is not clear. Was there a negative effect of contraception on fertility or not? This must be answered.

Recommendation stating “the need for counseling of clients by health provider” is not from of this research findings. Better to have conclusion based on the research findings.

Introduction

• The terms, Delayed fertility and infertility are confused. Better to have standard or operational definition of this terms as they are quite different.

Methods

• The authors described that they have selected the study subjects by SRS, what was the sampling frame? This is health facility study and the subjects usually be enrolled consecutively until the sample size is achieved

• My question on the definition of delayed fertility is answered here, however, it is not similar with infertility and the term infertility or delayed fertility must not be used interchangeably.

• How did you confirm that all are pregnant? At what gestational age did you include them? if majority are in the second trimester you might have excluded women who were pregnant and had abortions in the first trimester, which occurs at least in 15% of the cases. Or you have to put this as a limitation of this study.

Result

• Where is the age of the participants? Age is a very crucial variable to talk about fertility. Better to show the age distribution of the women especially for those with delayed fertility, because age is a factor for fertility return.

• The frequency of sexual intercourse, in the part of others must be reclassified as these are many. Because the women with delayed fertility most likely be in this group.

• Gestational age must have been included in the obstetrics variables of the study participants to know the proportion of trimesters. If majority are in the second trimester and above, there a possibility to miss those ladies who had abortion and couldn’t be found at ANC which affects the overall proportion of women had return in fertility.

• Thus, all the possible confounders of return in fertility other than contraception must have been dealt with.

Discussion

• It is very brief and more literatures need to be used.

• Age was found to be a significant factor to affect fertility. Age above which limit? <18? >35? A unity increase in age above what? This has to be clear. Because it is known that early age and late age is associated with anovulation and late age and there may be a deal in fertility normally irrespective of contraception use.

• All the limitations should be stated.

Conclusion and recommendation

• The conclusion and recommendation have to be clear. What is the general conclusion? is the fertility return High or low? What will be the recommendation? Did the contraception use affect the return in fertility differently in your study area than others? What message do you convey to the public?

Reviewer #3: I have uploaded more extensive comments.

However, this article needs to be edited substantially for clarity. There are a lot of phrases that aren’t entirely clear and many details that are not included or are not elaborated upon sufficiently. The methods seem reasonable overall, but lack detail relevant to fully understand. The researchers seem to make a variety of assumptions regarding the data that are not fully explained.

6. PLOS authors have the option to publish the peer review history of their article (what does this mean?). If published, this will include your full peer review and any attached files.

Reviewer #1: **Yes: **Endalkachew Mekonnen Assefa

Reviewer #2: No

Reviewer #3: No

---

## [Author Response · Author response to Decision Letter 0]

12 Apr 2022

Janet E Rosenbaum, Ph.D

RE: Submission ID PONE-D-21-25440 R1 (Fertility return after contraceptive discontinuation and associated factors among women attended Family Guidance Association of Dessie model clinic, Northeast Ethiopia: a cross-sectional study)

Dear Dr. Janet E Rosenbaum,

Thank you very much for your email and the comments/suggestions of the reviewers and academic editor. We have looked at the comments and have revised our paper accordingly. We hope our paper improved as a result of incorporating the reviewers' and academic editor's comments and suggestions.

Please find for your kind consideration the following:

A rebuttal letter that responds to each point raised by the academic editor and reviewer(s). You should upload this letter as a separate file labeled 'Response to Reviewers'.

A marked-up copy of your manuscript that highlights changes made to the original version. You should upload this as a separate file labeled 'Revised Manuscript with Track Changes'.

An unmarked version of your revised paper without tracked changes. You should upload this as a separate file labeled 'Manuscript'.

While hoping that these changes would meet with your favourable consideration, we are happy to hear if there are more comments and suggestions. Please do not hesitate to let us know if you have any questions.

Yours Sincerely,

Yitayish Damtie 

School of Public Health, Wollo University 

Dessie, Ethiopia

Tel:+251943517982

E-mail: yitutile@gmail.com

Point by point response

Reviewer Comments

Reviewer 1

Introduction:

Comment 1: The first paragraph is unnecessary (It talks about infertility and....)

Response: Thank you dear reviewer. We have amended it.

Comment 2: The fourth paragraph is unnecessary...The introduction should focus on fertility return after contraceptive discontinuation and associated factors...But I didn't see about associated factors.

Response: Thank you. The comment is accepted and addressed accordingly. You can find the associated factors in the introduction section line 71-74 

Method:

Comment 3: How do you control reputation of cases? The same patient might came twice per week or weekly.

Response: Thank you for your constructive comment. It is true that the same patient might came twice per week or weekly. We put special markings on all study participants card immediately after data collection to avoid unnecessary collection of data from cases with repeated visit during the study period. We have tried to put this in the method section line 113-114.

Comment 4: I wonder that you get more than 400 pregnant mothers who fulfills the study criteria within a month (even less than 22 days in one clinic). I have a question on the reliability of the data.

Response: Thank you for your constructive comment. During research proposal development, we have tried to collect preliminary information regarding the number of pregnant women attended ANC unit of FGA of Dessie model clinic in 2018(a year before the study was conducted). As the 2018 report showed, more than 900 pregnant women on average attended ANC unit each month. So this is more than enough to get 402 pregnant women who fulfill the inclusion criteria within a month. In the real scenario most pregnant women including me prefer to receive ANC service from FGA dessie model clinic. Due to this the client flow is high in this clinic. 

Comment 5: Line 90: “…….. From previous similar works of literature” Was it validated? Since, u said this is the first study in Ethiopia. 

Response: Thank you for your constructive comment. We have checked the content and face validity of the tool by experts that have knowledge and research experience related to the subject matter.

Comment 6: There were ampoule studies related to your title in Ethiopia... please search more. E.g. [https://pubmed.ncbi.nlm.nih.gov/8698014/)

Response: Thank you. We have tried to modify the document by searching more literatures related to our title. 

Comment 7: This study had a lot of confounders especially memory related, sexually behavior to get pregnant. [They may discontinued due to side effects or other different reasons.] What was done to minimize thus confounders?

Response: Thank you dear reviewer for your constructive comment. To minimize memory related bias (recall bias), we have tried to design questioner carefully, explain questions in detail and provide adequate time for each study participants for better memorization. However, since recall bias cannot be eliminated in observational studies, we have tried to acknowledge it in the limitations of our study. In addition we have tried to conduct multivariable logistic regression analysis to control the effect of confounding variables.

Comment 8: What does mean 'unit increase in age...?’

Response: Thank you. We have try to modify it accordingly. 

Comment 9: What is your base to define delayed fertility after 12 months? Or operational definition?

Response: Thank you dear reviewer. It is just operational definition.

Discussion:

Comment 10. Discussion is superficial

Response: Thank you. We have tried to modify the discussion section by searching more literatures related to our title

Comment 11: Line 208-209: (Moreover, as age increases, the remaining eggs in older women are more likely to have abnormal chromosomes. What does it mean?

Thank you for your comment. To make it clear, as the age of the women increase, the number and quality of eggs decreases naturally and progressively from time to time until she reaches menopause causing subfertility. In addition, older age women more likely to have chromosomal problems in their mature eggs. This is because errors in meiosis are more likely to occur as a result of the aging process. Meiosis is the process in which sex cells divide and create new sex cells with half the number of chromosomes. Normally, meiosis causes each parent to give 23 chromosomes to a pregnancy. When a sperm fertilizes an egg, the union leads to a baby with 46 chromosomes. But if meiosis doesn’t happen normally, a baby may have abnormal chromosome or in other words, a baby may have an extra chromosome (trisomy), or have a missing chromosome (monosomy). These problems can cause early pregnancy loss even before the mother notice it.

Conclusion:

Comment 12: Is so elementary...should be improved.

Response: Thank you dear reviewer. We have tried revise the conclusion section accordingly. 

Reviewer 2

Title: 

Comment 1: The title “Fertility return after contraceptive discontinuation and associated factors

among women” is about fertility return after discontinuation, however, what I presented in the result is proportion if delayed fertility, which is the other way around. So it is better described as per the research topic and the objectives.

Response: Thank you dear reviewer. We have try to modify the result section accordingly. 

Abstract 

Comment 2: Line 28: which data was obtained from review of clients' records? Is interview not enough?

Response: Thank you dear reviewer for your important comment. Almost all the women’s data were obtained through face-to-face interview. However for those women who forgotten their last normal menstrual period, information on last normal menstrual period was obtained by reviewing client records.

Comment 3: Result: the phrase “Uterine contraceptive device has to be replaced by intrauterine contraceptive device”

Response: Thank you for your important comment. The comment is accepted and addressed accordingly in the abstract section line 35. 

Comment 4: The percentages have to be out of the parenthesis.

Response: Thank you. We have amended it. 

Comment 5: The conclusion is not clear. Was there a negative effect of contraception on fertility or not? This must be answered.

Response: Thank you. We have tried to modify the conclusion accordingly. 

Comment 6: Recommendation stating “the need for counseling of clients by health provider” is not from of this research findings. Better to have conclusion based on the research findings.

Response: Thank you dear reviewer for your important comment. We have amended the recommendation accordingly. 

Introduction: 

Comment 7: The terms, Delayed fertility and infertility are confused. Better to have standard or operational definition of this terms as they are quite different. Infertility is defined when a couple is unable to conceive in 12 months with unprotected adequate sexual intercourse. And usually it may have an obvious cause and it needs treatment to achieve conception. However, delayed fertility is subfertility,

Response: Thank you dear reviewer. We have amended the introduction part accordingly.

Method: 

Comment 8: The authors described that they have selected the study subjects by SRS, what was the sampling frame? This is health facility study and the subjects usually be enrolled consecutively until the sample size is achieved. 

Response: Thank you for your constructive comment. To make it brief, systematic random sampling technique was used to select study participants rather than purposive sampling. As the 2018 data showed, on average, a total of 936 pregnant women attended ANC room of FGA Dessie model clinic in every month. Using this information, interval (k) was determined by dividing the average client flow per month (N) to the total sample size (n) i.e. k=N/n, K=936/423, K=2.2≈2. So, study participants were selected in every two women until reaching the final sample size. We have amended the study population as “all systematically selected pregnant women attending ANC room of FGA Dessie Model Clinic during the study period”.

Comment 9: My question on the definition of delayed fertility is answered here, however, it is not similar with infertility and the term infertility or delayed fertility must not be used interchangeably.

Response: Thank you. The comment is accepted and addressed accordingly.

Comment 10: How did you confirm that all are pregnant? At what gestational age did you include them? If majority are in the second trimester you might have excluded women who were pregnant and had abortions in the first trimester, which occurs at least in 15% of the cases. Or you have to put this as a limitation of this study.

Response: Thank you for your important comment. This study was conducted among women who have already been registered as pregnant and started receiving antenatal care in antenatal care room of Family Guidance Association of Dessie model clinic. In this study, all pregnant women irrespective of their gestational age were included. Since the gestational age of the current pregnancy has no effect on fertility return, all women who are in the first, second and third trimester were eligible and included in this study. Pregnancy status was the main requirement for this study. Whenever they are confirmed pregnant at the time of data collection, they are eligible for the study. Pregnant women who experienced abortion cannot attend antenatal care room rather they attend abortion or delivery room so that they are by default excluded due to this reason. 

Result: 

Comment 11: Table 1: Where is the age of the participants? Age is a very crucial variable to talk about fertility. Better to show the age distribution of the women, because age is a factor for fertility return.

Response: Thank you dear reviewer for your important comment. We have tried to show the age distribution women on table 1 of our revised manuscript accordingly.

Comment 12: The frequency of sexual intercourse, in the part of others must be reclassified as these are many. Because the women with delayed fertility most likely be in this group.

Response: Thank you. The comment is accepted and addressed accordingly in table 2 of our revised manuscript.

Comment 13: Gestational age must have been included in the obstetrics variables of the study participants to know the proportion of trimesters. If majority are in the second trimester and above, there a possibility to miss those ladies who had abortion and couldn’t be found at ANC which affects the overall proportion of women had return in fertility.

Response: Thank you. We did not assess the gestational age of the women since the gestational age of the current pregnancy it has no effect on fertility return.

Comment 15: Line 162-163: “The proportion of delayed fertility return experienced by pregnant women after discontinuation of any modern contraceptive methods before the current pregnancy was 11.4 % (95% CI; (8-14.4%)).” Here the author must talk about fertility return which is 88.6% not the delay.

Response: Thank you. We acknowledge the problem and amended it accordingly in the result section line188-192. 

Comment 16: Line 164-166: this shows that even the fertility rate is very good as 75% conceived within 12 months which is a bit higher that we know from literatures, 70% for DMPA

Response: Thank you. We have amended the conclusion accordingly.

Comment 17: Line 174-175: “a unit increase in age…” was linear regression used? If it is Multivariable why a unit increase is used? And age above what? Or above which age limit?

Response: No, multivariable binary logistic regression was fitted to identify factors associated with delayed fertility return. By the way, continuous variables can be fitted in multivariable logistic regression without categorizing them. However, we have tried to categorize the age of respondents to know which age group is at high risk of having delayed fertility return in our revised manuscript. Accordingly, those aged greater than or equal to 35 years are at high risk of having delayed return in fertility. 

Comment 18: Table 4: which age group is at risk of having delayed return in fertility? Because age in an important variable for woman fertility than any other. What is the mean and median age of the women with delayed fertility?

Response: Thank you for your constrictive comment. According to this study, women aged ≥35 years are at risk of having delayed return in fertility as compared to women aged 15-24 years. The mean age of the women with delayed fertility was 29.5 years with SD ± 4.3 years whereas the median age of the women with delayed fertility was 28.5 years with IQR of 4 years. 

Discussion 

Comment 19: It is very brief and more literatures need to be used.

Response: Thank you for your important comment. We have tried to modify it by searching more literatures. 

Comment 20: Age was found to be a significant factor to affect fertility. Age above which limit? <18? >35? A unity increase in age above what? This has to be clear. Because it is known that early age and late age is associated with anovulation and late age and there may be a deal in fertility normally irrespective of contraception use.

Response: Thank you dear reviewer for your important comment. We have tried to categorize respondent’s age. Accordingly, those women aged ≥35 years were more likely experienced delayed fertility return as compared to pregnant women aged 15-24 years.

Comment 21: All the limitations should be stated.

Response: Thank you have tried to put the limitation of the study on page line 

Conclusion: 

Comment 22: The conclusion and recommendation have to be clear. What is the general conclusion? Is the fertility return high or low? What will be the recommendation? Did the contraception use affect the return in fertility differently in your study area than others? What message do you convey to the public?

Response: Thank you dear reviewer. We have amended conclusion and recommendation section accordingly.

Comment 23: Line 225-226: It is better to talk about the return in fertility than the delay. It is also better to describe that the return in fertility is high in Long-term and reversible contraception (DMPA, IUCD)...which will have a positive impact for the users

Response: Thank you. The comment is accepted and addressed accordingly.

Comment 24: 227-228: Did the authors assess whether the health providers did or didn't counsel the woman about the delay in fertility return when they chose DMPA? This is unfound in the result.

Why? The return in fertility is 75% which is very good. No base for this recommendation.

Response: Thank you for your comment. We did not assess whether or not the health care providers counsel women about the delay in fertility return when they chose DMPA. However in research, it is mandatory to provide recommendation for those concerned stakeholders about the main findings of a study. One of the main findings of our study was the association between using DMPA and delay in fertility return. Although the return in fertility among DMPA users was good (75%), the result of multivariable analysis indicated that pregnant women who used DMPA before the current pregnancy were 4.8 times more likely to experience fertility delay upon discontinuation as compared to women who used IUCD, OCP and implants. Based on this finding, it is mandatory to recommend health care providers to provide detail counseling about the advantage, disadvantage and the side effects (including its effect on fertility return) of each contraceptive methods for the users so that the users can freely choose the best one for themselves . 

Reference 

Comment 25: Check the references. The journals are not cited appropriately. 

Response: Thank you dear reviewer. We have tried to revise the reference accordingly.

Reviewer 3

Comment 1: The article needs to be edited for clarity. There are a lot of phrases that aren’t entirely clear and many details that are not included or are not elaborated upon sufficiently. 

Response: Thank you for your important comment. The whole part of the manuscript was revised. In addition, we have tried to put the detail for things that need further elaboration. 

Introduction

Comment 2: The introduction section needs to more fully interrogate the existing evidence. The citations indicate several strong articles on the topic of return to fertility after contraceptive use; however, the introduction doesn’t summarize the science in a robust fashion. 

Response: Thank you for your suggestion. We have tried to revise the introduction section accordingly. 

Comment 3: The paragraph on page 3 (lines 54-56) is quite extreme. The association between impaired fertility and murder are not likely as strong as those between impaired fertility and mental health disorders. This paragraph should be revised to indicate that certain associations of impaired fertility are rare compared to others. 

Response: Thank you. We have amended it accordingly. 

Material and Methods

Comment 4: Please provide background on FGA Dessie Model Clinic. It’s not clear what this is, who it serves, etc. Where is it? What context? What sorts of people go to this clinic- urban/peri-urban/rural? Stable population or migrating population? Did you collect data at first ANC visit or later ANC visit? 

Response: Thank you for your constructive comments. We have tried to provide background information on FGA Dessie Model Clinic in the method section page 4-5 line 83 to 91. 

Comment 5: It would be useful to know more about the questionnaire- how many questions? Were the responses open-ended or categorized? If questionnaire derived from previous questionnaires from similar works of literature, it would be useful to cite them.

Response: Thank you dear reviewer for your important comments. A total of 33 socio-demographic, behavioral, contraceptive, obstetric and disease related questions were used to assess fertility return and associated factors among pregnant women attending FGA model clinic of Dessie town. Most of the questions were closed ended questions. However there are some questions with open ended response. These open-ended response questions were categorized based on standards and using previous works of literatures. The questionnaire was derived from previous questionnaires from similar works of literature, we have tried to cite them in our revised manuscript.

Results

Comment 6: It is unclear to me how decisions were made around categorizing Socio-demographic data, notably non-binary behavioral characteristics. I would like confirmation that nearly 18% of the study reports a frequency of sexual intercourse of three times per day- this seems questionable and could skew the results given the association between frequency of intercourse and fertility.

Response: Thank you for your constrictive comment. Non-binary behavioral characteristics were categorized based on standards and based the results of previous literatures. In this study, 17.7% of the pregnant women reports a frequency of sexual intercourse of three times per day before they become pregnant. However, since it is the finding of the study, we have no chance without accepting it as it is. 

Comment 7: Please define further the variables. What is history of abortion- are you including spontaneous miscarriages in this definition? What is an irregular menstrual cycle? What would constitute history of medical illness- is it only TB and diabetes or inclusive of other illness? I’m curious about your cutoff for number of births, given that it would be interesting to better understand the distinction between those for whom this is their first pregnancy and those for whom its their second or greater- this is likely related to the sample, but bears noting. 

Response: Thank you. In this study, abortion was defined if the pregnancy was terminated before 28 weeks of gestation. It includes both spontaneous and induced. Similarly, a woman said to have irregular menstrual cycle if her cycle is shorter than 21 days or longer than 35 days whereas regular menstrual cycle is a cycle that ranges from 21 to 35 days. History of medical illness constitute only TB and diabetes. These two disease are public health important disease that influence women’s fertility. Evidences suggested that 10 up to 15% of infertility cases among women are as result of TB infection that spread to the reproductive organs (uterus, ovaries and the Fallopian tubes) through bloodstream. Diabetes also plays a significant role in reproductive disorders such as anovulation, menstrual disorders and infertility as it causes obesity. We have re-categorized the cutoff for number of births as “never give birth”, “one birth” and ≥2 births. We have tried to put the operational definition in the method section line 125-131.

Discussion

Comment 8: When citing previous studies, please note the authors, not only the location of the study. 

Response: Thank you. It is difficult to put authors for all studies. However, efforts were made to put the authors for studies used in the discussion section accordingly.

Comment 9: Increasing age is associated with decreasing fertility. This is steeped in research. Please cite research on page 16 paragraph on lines 203-209. It currently reads in a speculative manner; however, in reality, these associations are real and documented.

Response: Thank you dear reviewer. We have tried to cite research for it accordingly in the discussion section line 239.

---

## [Decision Letter · Decision Letter 1]

1 Dec 2022

PONE-D-21-25440R1Fertility return after contraceptive discontinuation and associated factors among women attended Family Guidance Association of Ethiopia Dessie model clinic, Northeast Ethiopia: a cross-sectional studyPLOS ONE

Dear Dr. Damtie,

Thank you for submitting your manuscript to PLOS ONE. After careful consideration, we feel that it has merit but does not fully meet PLOS ONE’s publication criteria as it currently stands. Therefore, we invite you to submit a revised version of the manuscript that addresses the points raised during the review process.

We look forward to receiving your revised manuscript.

Kind regards,

Janet E Rosenbaum, Ph.D.

Academic Editor

PLOS ONE

Reviewers' comments:

Reviewer's Responses to Questions

**Comments to the Author**

1. If the authors have adequately addressed your comments raised in a previous round of review and you feel that this manuscript is now acceptable for publication, you may indicate that here to bypass the “Comments to the Author” section, enter your conflict of interest statement in the “Confidential to Editor” section, and submit your "Accept" recommendation.

Reviewer #4: (No Response)

Reviewer #5: (No Response)

2. Is the manuscript technically sound, and do the data support the conclusions?

Reviewer #4: Yes

Reviewer #5: Partly

3. Has the statistical analysis been performed appropriately and rigorously? 

Reviewer #4: Yes

Reviewer #5: Yes

4. Have the authors made all data underlying the findings in their manuscript fully available?

Reviewer #4: Yes

Reviewer #5: Yes

5. Is the manuscript presented in an intelligible fashion and written in standard English?

Reviewer #4: Yes

Reviewer #5: Yes

6. Review Comments to the Author

Reviewer #4: This work is interesting but data reported didn't give new informations about fertility return after contraceptive discontinuation.

This fiel has been largely studied and data reported in this work didn't add new informations (1).

1 Girum T, Wasie A. Return of fertility after discontinuation of contraception: a systematic review and meta-analysis. Contracept Reprod Med. 2018 Jul 23;3:9. doi: 10.1186/s40834-018-0064-y. PMID: 30062044; PMCID: PMC6055351.

Reviewer #5: Reviewer comments:

Title: Fertility return after contraceptive discontinuation and associated factors among women attended Family Guidance Association of Dessie model clinic, Northeast Ethiopia: a cross-sectional study

1. Abstract: : Women who use hormonal contraception…..: you have used hormonal contraception the reader will understand you have checked for only hormonal ones !!! Use it at the title also and be sure that when you are recording on the contraception within the text to be hormonal.

2. Abstract line 36: OCP includes combined oral contraception piles and progesterone only piles!!! please refer to it in a more definite way in all the text

3. Introduction, first line: Globally, around 15% of couples experience infertility!!! That is true; the proportion of delayed fertility return among currently pregnant women after discontinuation contraceptive methods was 11.4%in your research!! Do you think that there was any delay in getting pregnant? almost 88% became pregnant within 1 year of attempting pregnancy

4. Introduction line 57-58: These include socio-demographic factors (age) [16]: socio-demographic factors does not means (age ) to put the word between bracket !! either add other factors with the age and this will need many references or delete it

5. Introduction( MMR,FGA) are two abbreviations needs to be proceeded by full description

6. Introduction line 67: . One of the reasons for not using contraceptive methods might be due to fertility delay associated with contraceptive: At introduction you do not add your suggestions for a result!! You may shift it to the discussion section.

7. Page 4 sample size: The sample size was determined by using single population proportion formula by considering the proportion of delayed fertility return as 50%

a. Using 50% reference is a very high percent!! Although there was no research done in this community , in these situations you may use the rate in nearby countries or even any published article on the same purpose

b. A cross sectional study means you will survey in a population about any one whom have used contraception and failed to get pregnant within one year so ideally does not need sample size estimation being a cross sectional study

c. Using single population proportion: which population due mean? Was it at antenatal care unites? Or gynecology outpatient clinic? Or may infertility unit?

8. Result: when I reached the Socio-demographic characteristics then understood that your participants were already pregnant now!!! And still didn’t get the setting of your research!!!!!

9. Page 7, line 131: Ethiopian birr is a local currency which is not understandable for me and the readers!!! Please add to it how much it correspond to using a global currency like American dollar

10. Page 8, line 141: chew chat!! I didn’t understand what it means!! please clarify it in your methods section

11. Page 9, line 144: Pregnant women perform sexual intercourse three times a day and three times a week respectively: what do you mean by this information? They are already pregnant and having intercourse many times does not give you any information about your objectives!!!

12. Page 10, line 156: Four (1%) and 6(1.5%) of pregnant women had a history of Sexual Transmitted Infection (STI) and other medical illness (tuberculosis and diabetic Malthus) :

Although the delayed in getting pregnancy in your sample size was 11% (less than the rate of infertility all over the world (15%), still you have-not excluded whom already infertile from other causes like sexually transmitted disease and uncontrolled diabetes mellitus!! you have to clarify this point or to remove the 4 ladies having other causes for infertility not delay in getting pregnancy after withdrawal of contraception.

7. PLOS authors have the option to publish the peer review history of their article (what does this mean?). If published, this will include your full peer review and any attached files.

Reviewer #4: No

Reviewer #5: **Yes: **PROFESSOR SHAHLA KAREEM ALALAF

---

## [Author Response · Author response to Decision Letter 1]

8 May 2023

Janet E Rosenbaum, Ph.D

RE: Submission ID PONE-D-21-25440 R2 (Fertility return after contraceptive discontinuation and associated factors among women attended Family Guidance Association of Dessie model clinic, Northeast Ethiopia: a cross-sectional study)

Dear Dr. Janet E Rosenbaum,

Thank you very much for your email and the comments/suggestions of the reviewers and academic editor. We have looked at the comments and have revised our paper accordingly. We hope our paper improved as a result of incorporating the reviewers' and academic editor's comments and suggestions.

Please find for your kind consideration the following:

A rebuttal letter that responds to each point raised by the academic editor and reviewer(s). You should upload this letter as a separate file labeled 'Response to Reviewers'.

A marked-up copy of your manuscript that highlights changes made to the original version. You should upload this as a separate file labeled 'Revised Manuscript with Track Changes'.

An unmarked version of your revised paper without tracked changes. You should upload this as a separate file labeled 'Manuscript'.

While hoping that these changes would meet with your favourable consideration, we are happy to hear if there are more comments and suggestions. Please do not hesitate to let us know if you have any questions.

Yours Sincerely,

Yitayish Damtie 

School of Public Health, Wollo University 

Dessie, Ethiopia

Tel:+251943517982

E-mail: yitutile@gmail.com

Point by point responses

Reviewer #4: This work is interesting but data reported didn't give new information’s about fertility return after contraceptive discontinuation. This field has been largely studied and data reported in this work didn't add new information’s (1)

1 Girum T, Wasie A. Return of fertility after discontinuation of contraception: a systematic review and meta-analysis. Contraception Reprod Med. 2018 Jul 23;3:9. doi: 10.1186/s40834-018-0064-y. PMID: 30062044; PMCID: PMC6055351.

Response: Thank you dear reviewer for your comment. Even if return of fertility after discontinuation of contraception has been largely studied and a systematic review and meta-analysis was done at the global level, the field has not been studied yet in Ethiopia. So, this study would have a paramount significance in providing important information for concerned individuals who are working to improve the reproductive health of Ethiopian women. 

Reviewer #5: 

Title: Fertility return after contraceptive discontinuation and associated factors among women attended Family Guidance Association of Dessie model clinic, Northeast Ethiopia: a cross-sectional study

1. Abstract: Women who use hormonal contraception…... you have used hormonal contraception the reader will understand you have checked for only hormonal ones!!! Use it at the title also and be sure that when you are recording on the contraception within the text to be hormonal.

Response: Thank you dear reviewer for your constructive comment. We have amended it. 

2. Abstract line 36: OCP includes combined oral contraception piles and progesterone only piles!!! please refer to it in a more definite way in all the text.

Response: Thank you. As it is indicated in the abstract line 35 and throughout the manuscript, we used the abbreviation OCP to refer oral contraceptive pills which includes both combined oral contraception piles and progesterone only piles. 

3. Introduction, first line: Globally, around 15% of couples experience infertility!!! That is true; the proportion of delayed fertility return among currently pregnant women after a discontinuation contraceptive method was 11.4%in your research!! Do you think that there was any delay in getting pregnant? Almost 88% became pregnant within 1 year of attempting pregnancy.

Response: Definitely! Although the figure is not high, 11.4% of currently pregnant women experienced a delayed fertility return after discontinuation of contraceptive methods before they became pregnant. 

4. Introduction line 57-58: These include socio-demographic factors (age) [16]: socio-demographic factors does not means (age ) to put the word between bracket !! either add other factors with the age and this will need many references or delete it

Response: Many thanks for your comment. We have amended it. 

5. Introduction (MMR, FGA) are two abbreviations needs to be proceeded by full description

Response: Thank you for your comment. The full description of FGAE has already written in the abstract section so that we can use the abbreviated form in the subsequent sections. Regarding the abbreviation MMR, we removed it from the introduction due to other reviewer suggestion.

6. Introduction line 67: One of the reasons for not using contraceptive methods might be due to fertility delay associated with contraceptive: At introduction you do not add your suggestions for a result!! You may shift it to the discussion section.

Response: Thank you for your suggestion. We have revised it accordingly. 

7. Page 4 sample size: The sample size was determined by using single population proportion formula by considering the proportion of delayed fertility return as 50%

A. Using 50% reference is a very high percent!! Although there was no research done in this community , in these situations you may use the rate in nearby countries or even any published article on the same purpose

Response: Thank you for your important comment. Many scholars recommended using 50% reference for sample size calculation in the case of unknown prevalence since it provides maximum sample size. It is not recommended to use the rate in nearby countries for sample size calculation due to the variation in the concern of government on health and health related issues and health care interventions across countries. In addition, the sample size calculated from studies conducted in nearby countries may not be larger than the sample size calculated using 50% reference for sample size since it provides maximum sample size. The power of the study and precision of the estimate increases with an increase in sample size. As long as it doesn’t affect the power and the precision of the study, it is better to use 50% reference in case of unknown prevalence. 

B. A cross sectional study means you will survey in a population about any one whom have used contraception and failed to get pregnant within one year so ideally does not need sample size estimation being a cross sectional study.

Response: Sample size estimation is mandatory for any type of quantitative study including cross-sectional study. This is because if we survey a small number of women, we will not be able to detect an effect and may produce inconclusive result. On the other hand, if we survey large number of women, it may waste scares resources (time, money, manpower etc...). So, determination of a sample size which is an appropriate is crucial to avoid the aforementioned issues. 

C. Using single population proportion: which population due mean? Was it at antenatal care unites? Or gynecology outpatient clinic? Or may infertility unit?

Response: Thank you for your comment. It is to mean antenatal care unites. 

8. Result: when I reached the Socio-demographic characteristics then understood that your participants were already pregnant now!!! And still didn’t get the setting of your research!!!!!

Response: Really! Our study participants were women who are pregnant at the time of study. The aim of our study was to assess whether fertility was returned (whether women became pregnant) within 12 months of contraceptive discontinuation or not and to determine factors for those women who experience delayed in return of fertility after contraceptive discontinuation. Fertility return was defined if conception occurred within 12 months following discontinuation of any modern contraceptive methods. On the other hand, delayed fertility return was defined if conception occurred after 12 months following discontinuation of any modern contraceptive methods. So to achieve our objective, study participants should be pregnant women. 

9. Page 7, line 131: Ethiopian birr is a local currency which is not understandable for me and the readers!!! Please add to it how much it correspond to using a global currency like American dollar

Response: Thank you for your important comment. We have tried to change it to American dollar.

10. Page 8, line 141: chew khat!! I didn’t understand what it means!! Please clarify it in your methods section.

Response: Thank you. To make it clear, Khat is a fresh green leave plant native to eastern Africa and the Arabian Peninsula. In 1980, WHO classified it as an illicit substance due to the potential for psychological dependence. Khat contains a psycho-active ingredient cathinone which is said to cause excitement, and euphoria. Its regular consumption negatively impacts the human central nervous system, systemic blood pressure, psychological health and reproductive system causing reproductive toxicity and sexual dysfunction. We have tried to clarify it in the method section as well.

11. Page 9, line 144: Pregnant women perform sexual intercourse three times a day and three times a week respectively: what do you mean by this information? They are already pregnant and having intercourse many times does not give you any information about your objectives!!!

Response: Yes, this study was conducted among women who are already pregnant. However, we used frequency of sexual intercourse as a factor to assess how often they performed sexual intercourse before they become pregnant since frequency of sexual intercourse is an important factor for the occurrence of pregnancy (fertility return). It is known that women who perform sexual intercourse three times a day, three times a week, three times a month and three times a year have no equal chance of becoming pregnant.

12. Page 10, line 156: Four (1%) and 6(1.5%) of pregnant women had a history of Sexual Transmitted Infection (STI) and other medical illness (tuberculosis and diabetic Malthus). Although the delayed in getting pregnancy in your sample size was 11% (less than the rate of infertility all over the world (15%), still you have-not excluded whom already infertile from other causes like sexually transmitted disease and uncontrolled diabetes mellitus!! you have to clarify this point or to remove the 4 ladies having other causes for infertility not delay in getting pregnancy after withdrawal of contraception.

Response: Thank you for your comment. Our study participants were women who are pregnant at the time of study. Infertility could not be an issue for our study participants since they are already pregnant. Studies suggested that having STI and other medical illness like Tuberculosis and diabetic Malthus can delay pregnancy or may causes infertility if they are left untreated. Due to these, we want to test their statistical significance association with delay in getting pregnancy.

---

## [Decision Letter · Decision Letter 2]

6 Jun 2023

Fertility return after hormonal contraceptive discontinuation and associated factors among women attended Family Guidance Association of Ethiopia Dessie model clinic, Northeast Ethiopia: a cross-sectional study

PONE-D-21-25440R2

Dear Dr. Damtie,

We’re pleased to inform you that your manuscript has been judged scientifically suitable for publication and will be formally accepted for publication once it meets all outstanding technical requirements.

Kind regards,

Janet E Rosenbaum, Ph.D.

Academic Editor

PLOS ONE

Additional Editor Comments (optional):

Reviewers' comments:

Reviewer's Responses to Questions

**Comments to the Author**

1. If the authors have adequately addressed your comments raised in a previous round of review and you feel that this manuscript is now acceptable for publication, you may indicate that here to bypass the “Comments to the Author” section, enter your conflict of interest statement in the “Confidential to Editor” section, and submit your "Accept" recommendation.

Reviewer #5: All comments have been addressed

2. Is the manuscript technically sound, and do the data support the conclusions?

Reviewer #5: Yes

3. Has the statistical analysis been performed appropriately and rigorously? 

Reviewer #5: Yes

4. Have the authors made all data underlying the findings in their manuscript fully available?

Reviewer #5: Yes

5. Is the manuscript presented in an intelligible fashion and written in standard English?

Reviewer #5: Yes

6. Review Comments to the Author

Reviewer #5: The correspond author responded to the previous comments were adequately addressed. The article is suitable now for publication

7. PLOS authors have the option to publish the peer review history of their article (what does this mean?). If published, this will include your full peer review and any attached files.

Reviewer #5: **Yes: **SHAHLA KAREEM ALALAF

---

## [Editor Report · Acceptance letter]

3 Jul 2023

PONE-D-21-25440R2 

Fertility return after hormonal contraceptive discontinuation and associated factors among women attended Family Guidance Association of Ethiopia Dessie model clinic, Northeast Ethiopia: a cross-sectional study 

Dear Dr. Damtie:

I'm pleased to inform you that your manuscript has been deemed suitable for publication in PLOS ONE. Congratulations! Your manuscript is now with our production department. 

Kind regards, 

on behalf of

Dr. Janet E Rosenbaum 

Academic Editor

PLOS ONE